# Thermal Degradation of Organophosphorus Flame Retardants

**DOI:** 10.3390/polym14224929

**Published:** 2022-11-15

**Authors:** Bob A. Howell

**Affiliations:** Science of Advanced Materials, Center for Applications in Polymer Science, Department of Chemistry and Biochemistry, Central Michigan University, Mt. Pleasant, MI 48859-0001, USA; bob.a.howell@cmich.edu

**Keywords:** phosphorus esters, degradative elimination, oxygenation level at phosphorus, polymer combustion, flammability inhibition, radical fragmentation, flame retardant synthesis

## Abstract

The development of new organophosphorus flame retardants for polymeric materials is spurred by relatively low toxicity, effectiveness, and demand for replacement of more traditional materials. To function, these compounds must decompose in a degrading polymer matrix to form species which promote modification of the solid phase or generate active radical moieties that escape to the gas phase and interrupt combustion propagating reactions. An understanding of the decomposition process for these compounds may provide insight into the nature of flame retardant action which they may offer and suggest parameters for the synthesis of effective new organophosphorus flame retardants. The thermal degradation of a series of organophosphorus esters varying in the level of oxygenation at phosphorus—alkyl phosphate, aryl phosphate, phosphonate, phosphinate—has been examined. Initial degradation in all cases corresponds to elimination of a phosphorus acid. However, the facility with which this occurs is strongly dependent on the level of oxygenation at phosphorus. For alkyl phosphates elimination occurs rapidly at relatively low temperature. The same process occurs at somewhat higher temperature for aryl phosphates. Elimination of a phosphorus acid from phosphonate or phosphinate occurs more slowly and at much higher temperature. Further, the acids formed from elimination rapidly degrade further to evolve volatile species.

## 1. Introduction

The development of polymeric materials beginning in the mid-20th century has been a boon to modern society [1,2,3,4,5]. The high standard of living enjoyed by most citizens of the developed world would not be possible in absence of polymeric materials. In general, polymers are flammable and for most applications must be flame retarted. Traditionally, organohalogen compounds have been popular flame retardants [6,7]. In particular, multiply brominated diphenyl ethers have been widely used. The most prominent has been decabromodiphenyl ether. These compounds are inexpensive (diphenyl ether is an industrial by-product), readily available and effective gas-phase flame retardants. In a degrading polymer matrix, they decompose to form hydrogen bromide which escapes to the gas phase to quench combustion propagating radicals. However, these materials tend to migrate from a polymer matrix into which they have been incorporated [7,8]. They have become widely distributed in the human environment and may be found in everything from house dust to food packaging. More importantly, when used items are discarded in a landfill these materials leach into the surroundings where they are stable, bioaccumulate, and may enter the human food chain [9,10]. Moreover, at high temperature, as in a fire, these materials are converted to volatile, very toxic dioxins and furans [11,12]. Human exposure to organohalogen flame retardants may lead to the development of a number of disease states most related to endocrine disruption [13,14]. For this reason, these compounds are coming under increasing regulatory and societal pressure worldwide. In recent years, there has been much activity directed toward the development of organophosphorus replacements for these materials [15]. In particular, organophosphorus flame retardants derived from renewable, nontoxic biosources are particularly attractive and are receiving increasing attention [16]. Flame retardants may be effective in the solid phase or the gas phase as illustrated by the simple diagram shown in Figure 1. When heated to sufficiently high temperature, most polymers begin to degrade to form small-molecule fragments which escape to the gas phase.

In the gas phase, these fragments mix with oxygen and if the temperature is sufficiently great, at or above the ignition temperature, combustion may be initiated. Combustion is comprised of a series of exothermic radical reactions. Heat and light (flame) and, depending on the efficiency of combustion, particulates (soot, smoke) are formed. Heat may feedback to the polymer surface to enhance the rate of polymer degradation (pyrolysis) and the consequent formation of fuel fragments to feed the combustion process [17,18]. There are two opportunities to interrupt this process. One is in the solid phase—to decrease the rate of fuel formation. Some flame retardants promote the formation of a char layer at the surface of the polymer to act as an insulating layer to inhibit heat feedback and limit the rate of pyrolysis and formation of volatile fuel fragments. Alternatively, a flame retardant additive may decompose in the degrading polymer matrix to form volatile radical species which escape to the gas phase and interrupt combustion propagation reactions. Organophosphorus compounds may promote flame retardant activity in either the solid phase or the gas phase depending largely on structure [19,20,21,22]. Most, those with a high level of oxygenation at phosphorus (i.e., phosphates), promote char formation in the solid phase. Far fewer, those with a low level of oxygenation at phosphorus, principally those containing the 9,10-dihydro-9-oxa-10-phosphaphenanthrene-10-oxide moiety (DOPO phosphonate), decompose in the degrading polymer matrix to generate volatile radical species which enter the gas phase and quench important combustion branching reactions [23,24]. Compounds containing both types of phosphorus functionality may display two types of flame-retardant activity [25]. In either case, the flame retardant additive must decompose in the degrading polymer matrix to generate species responsible for the observed flame retardant action. Ideally, a flame retardant additive should undergo decomposition near the temperature at which the polymer degrades. Insight into the mode of flame retardant action may be gained from a knowledge of additive thermal degradation.

## 2. Experimental

Materials, methods, and instrumental techniques have been described in detail previously [26,27,28,29]. Thermogravimetry was performed using a TA Instruments Q500 instrument. For dynamic analysis, a heating rate of 10 °C/min was used. Samples (8–10 mg) were contained in a platinum pan. The sample compartment was purged with dry nitrogen at 60 mL/min during analysis. Infrared spectra were obtained using attenuated total reflectance and a thermo scientific Nicolet 380 FI-IR spectrophotometer. Electroscopy (ESI) chemical ionization mass spectrometry was accomplished using a Waters micromass LCT Premier XE orthogonal acceleration time-of-flight (TOF) mass spectrometer. Nuclear magnetic resonance (NMR) spectra were obtained using a Varian Mercury 300 MH_z_ spectrometer and approximately 10% solutions in deuterochloroform. ^1^H and ^13^C chemical shifts are reported in parts-per-million (δ) with respect to the resonance for tetramethylsilane (TMS) as internal reference (δ = 0.00); ^31^P chemical shifts with respect to the resonance for triphenyl phosphate (δ = −18.00) as internal reference.

## 3. Results and Discussion

Four phosphorus esters of isosorbide were synthesized using either direct phosphorylation or the Atherton-Todd procedure [26]. Structures for these compounds are shown in Figure 2. One (IDEA) is a dialkyl phosphate, one [IDPA] is a diaryl phosphate, one (IDPO) is a phosphinate, and one (IDOPYL) is a phosphonate. Thus, these compounds represent phosphorus esters with a range of oxygenation (2–4 oxygen atoms) at phosphorus and containing both alkyl and aryl moieties in the functional group. The mode of thermal degradation of these compounds might reflect expected action as flame retardants.

The thermal degradation of these compounds has been documented using thermogravimetry. Changes occurring during degradation were recorded using infrared spectroscopy and components in the degradation char identified using ESI-MS or NMR spectroscopy. The thermal degradation profiles for these compounds are displayed in Figure 3 [28]. Several observations are possible. 

First, the diethyl ester (IDEA) is the least stable by a significant margin and degradation occurs in two steps with degradation onset temperature of 156 °C (from the corresponding derivative plot of mass loss versus temperature) [28]. The diphenyl ester (IDPA) is considerably more stable (degradation onset at 289 °C) but less stable than either the phosphonate (IDOPYL) or the phosphinate (IDPO) (degradation outset at 323 °C and 338 °C, respectively). Further, the residue remaining after degradation of the phosphates (13% of the initial sample mass for the diethyl ester, 24% for the diphenyl ester) is much greater than that for the degradation of either the phosphonate (12%) or the phosphinate (2%). Clearly these later two undergo decomposition, subsequent to the initial degradation, to volatilize much of the residue. The difference in degradation behavior for these compounds is more apparent in plots for isothermal degradation shown in Figure 4 [28].

The phosphate esters (IDEA, IDPA) degrade rapidly at this temperature, the phosphinate (IDPO) much less rapidly, and the phosphonate (IDOPYL) is little changed after several hours at this temperature. The initial degradation process for these compounds is apparent from infrared spectra recorded as decomposition occurs. For comparison two sets of spectra are shown below (Figure 5 and Figure 6). 

The rapid degradation of the diethyl ester is readily apparent from the rapid disappearance of C-H absorption near 3000 cm^−1^ (largely absent after only one hour at 260 °C) and the appearance of a broad band characteristic of a phosphorus acid in the same region. The other major change in the spectra is the appearance and growth of a band at 1644 cm^−1^ (alkene). These observations suggest that the major transformation occurring is elimination of a phosphorus acid to generate an alkene (a variant of a well-known method for the preparation of alkenes) [28,30].

From the mass loss observed in dynamic TGA, it would appear that initial elimination occurs in the ethyl groups (first degradation step) to liberate ethylene and then in the isosorbide unit. The degradation for the *bis*-phosphonate (DOPO ester) is similar but much slower. Bands for the C-H absorption are still present in the spectrum after six hours at 260 °C. However, the same process, elimination of a phosphorus acid to form an unsaturated compound, does occur albeit over a much longer time. The formation of the phosphorus acids was confirmed by extraction of the residue with acetone followed by analysis using ESI-MS. For three of the esters the corresponding phosphorus acid was prominently present. For the residue from decomposition of isosorbide *bis*-diethylphosphate only phosphoric acid was present. Comparable observations have been made for this series of esters from another substrate [30]. In this instance, the DOPO acid sublimed, was collected, and the structure confirmed using NMR spectroscopy. Most notably, the ^31^P spectrum contains a single sharp peak at δ 13.1. The mode of thermal degradation of this series of isosorbide *bis*-phosphorus esters is depicted in Figure 1. In each case, the initial process is elimination to form a known diene and a previously characterized phosphorus acid. Ethyl esters are particularly prone to elimination and extrude ethylene at relatively low temperature.

As noted in the TGA plots, the acids generated from the phosphinate and the phosphonate undergo further degradation to form volatile species. This most likely reflects the formation of the PO radical [23,24]. This process is outlined in Figure 2. While the PO radical may escape to the gas phase to capture combustion propagating radicals, the more reactive hydroxyl radical most likely reacts with species in the degrading polymer matrix, probably by hydrogen atom abstraction.

As flame retardant polymer additives, the behavior of organophosphorus esters is largely dependent on the level of oxygenation at phosphorus [27]. Those with high levels of oxygenation at phosphorus (i.e., phosphates) readily undergo decomposition in the degrading polymer matrix to generate a phosphorus acid. The presence of the acid promotes cationic crosslinking and char formation. A char layer at the surface of the polymer acts as an insulation barrier to inhibit heat feedback from the combustion zone. Pyrolytic generation of volatile fuel fragments is retarded and delivery of fuel to support combustion is limited. Thus, organophosphorus compounds with a high level of oxygenation at phosphorus are solid phase active. Compounds with a low level of oxygenation at phosphorus behave differently. They undergo initial degradation at high temperature to form oxygen-deficient phosphorus acids which undergo further decomposition to form volatile radical species which may escape to the gas phase. To be effective, these species must be capable of efficiently scavenging combustion propagating radicals. Evolution of a phosphorus-containing fragment is not sufficient for flame retardancy [24]. It must be an active species with radical-trapping capability. To date, only the PO radical has been rigorously identified as such a species, hence the popularity of DOPO derivatives as flame retardant additives. Compounds of this kind are more effective as flame retardant additives than are more highly oxygenated counterparts [28] This has implications for the synthesis of new compounds for use as flame retardant additives. New compounds should contain a low-level oxygenation at phosphorus and possess an accessible radical pathway for degradative fragmentation to form active volatile species, preferably leading to the formation of a thermodynamically stable product. Often the preparation of new additives is based on intuition or empirical observations, or combinations of agents to compensate for deficiencies of the primary flame retardant. A more rational approach might be the design of new structures with features to assure a high degree of effectiveness.

In the development of new organophosphorus flame retardant additives concern for toxicity and environmental contamination and persistence must be paramount. Compounds derived from nontoxic, renewable and abundant biomaterials offer great potential. Among the compounds described here, the *bis*-dopyloxy isosorbide derivative presents outstanding properties. It is easily prepared, compatible with a range of polymers, displays good thermal stability with a degradation onset of over 300 °C so can be used with polymers that process at relatively high temperature, and confers good flame retardancy to a polymer matrix into which it has been incorporated [27].

Oligomeric flame retardants offer the advantage of reduced migration over small molecule counterparts. Recently developed techniques permit the ready synthesis of hyperbranched poly(ester)s of precise structure from the abundantly available biomonomers, glycerol and adipic acid [31,32,33]. Using the Martin-Smith model for the determination of initial monomer ratios, polymerization may be carried out to high degrees of monomer conversion without gelation. The materials have well-defined molecular weight, structure and a single type of endgroup. The presence of a single kind of endgroup is important both for stability (these materials do not gel on storage) and ease of subsequent functionalization. Materials of a range of molecular weights may be generated in a simple one-step process. As poly(ester)s they are fully compatible with many polymeric materials. Hydroxyl endgroups may be readily converted to a variety of organophosphorus esters with the dopyloxy perhaps having the greatest attractiveness as a flame retardant additive. The size (molecular weight) of these materials may be adjusted via synthesis to meet a variety of use demands. If both gas phase and solid phase activity are desired, a mixture of polymers containing on the one hand dopyloxy endgroups and on the other phosphate (probably diphenyl phosphate) may be used. Because these materials are esters and have a hyperbranched structure, they display an outstanding plasticizing effect [34]. Thus, a single additive may provide both plasticization and flame retardancy. Methods of production are scalable, and these materials would seem to offer great potential for commercial exploitation.

## 4. Conclusions

Using isosorbide as a biobased substrate, the thermal degradation of a series of organophosphorus flame retardants has been explained. These represent a series of phosphorus esters of varied structure (varying levels of oxygenation at phosphorus—alkyl phosphate, aryl phosphate, phosphonate, phosphinate). All undergo initial degradation via elimination of a phosphorus acid. This is a very facile process for alkyl phosphates (multiple pathways for elimination). Aryl phosphates undergo decomposition via the same process but at a somewhat lower rate. Degradative elimination occurs much more readily for these compounds (high level of oxygenation at phosphorus) than for compounds containing low levels of oxygenation at phosphorus, phosphonate, and phosphinate. These compounds undergo initial elimination but more slowly and at much higher temperature than do phosphates. In these cases, the initially formed acids undergo thermal fragmentation to expel volatile species to the gas phase. These observations have implications for both the mode of flame retardant action for these compounds and for the development of new organophosphorus flame retardants.

## Data Availability

The data presented in this study are available on request from the corresponding author.

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
