# Peer review of "Thermal Degradation of Organophosphorus Flame Retardants"

_polymers, 2022, doi:10.3390/polym14224929_

Round 1

Reviewer 1 Report

The synthesis of novel organophosphorus compounds which can be suitable as alternative to replace bromine-containing flame retardants has been the subject of a huge number of studies during the last decades. However, to targeted design of very effective phosphorus-containing flame retardants,  detailed knowledge on their mode of action is urgently necessary. In that context, the pathway of decomposition of these compounds when exposed to heat is of high importance.

In the last few years, a couple of articles have been published by Howell et al. dealing with the synthesis of phosphorus-containing esters of isosorbide, testing their flame-retardant effect and investigations of the decomposition mechanism of these compounds. These studies confirmed results of other working groups that the number of oxygen atoms surrounding the phosphorus atom have crucial influence on thermal stability and the mode of action (gas phase or condensed phase).

The presented study of Howell is a supplement to these articles (ref. 26-29). The presented manuscript provides a few novel information. Here, investigations by means of infrared spectroscopy on the initial decomposition step as well as results of NMR spectroscopy and ESI-MS were presented. The investigations revealed that the initial decomposition step is the elimination of an alkene species with formation of a phosphorus-containing acid. However, the degradation process of the phosphorus-containing acids formed during the initial decomposition of the esters provokes some questions and should be discussed by the author. The author proposed the formation of a PO radical as well as an OH radical. In case it is true, it would have the following consequence: Whereas the PO radical is known to quench the chain reaction inside the flame, the OH radical is a very reactive species promoting the burning process!! That means, the decomposition would generate a species which can interrupt the burning process as well another one that is part of the burning process in the flame. In addition, it would have another consequence: phosphorus compounds which are not able to form an acid during decomposition would be better flame retardants (for instance: DOPO derivatives with a P-C bond instead C-O bond; it seems to me, that is really the case!). The author should urgently discuss this aspect before publishing the article draft!

Some additional remarks: The TGA curves given in Fig. 3 and 4, respectively, have already been published in a previous publication (it should be mentioned!). Please check and correct the references part thoroughly, because it contains some errors: first of all, please insert the page numbers in ref. 16. There is a lot of incorrect authors names (for instance: ref 19, please replace Doring by Doering or Döring, replace Alstadt by Altstädt, Pospieck by Pospiech and Hoffmen by Hoffmann; ref. 22, Ciesieloki by Ciesielski, Muller by Mueller or Müller ....).  

Reviewer 2 Report

Please Refer attachment

Reviewer 3 Report

Very good manuscript. Only the figures are of low quality:

Fig. 1: Poor quality, not straight lines

Fig. 2: Poor quality, no x-axis title

Fig. 5 & 6: Poor quality, not sufficient axes-titles

Scheme 1 should be better called Fig. 7: Explanations of the chemical structures are needed

Same counts for scheme 2

Round 2

Reviewer 1 Report

I recommend the second version of the manuscript for publishing in the journal Polymers.

The author inserted a few sentences regarding the role of the OH radicals which seem to be formed by the decomposition of the flame retardants.  He argues that the OH radicals remain in the condensed phase and therefore cannot participate in the chain reaction of the burning process.

However, the author should check the reference part thoroughly again, because it still contains some errors, for instance please ref. 23 by:

A. Schäfer, S. Seibold, W. Lohstroh, O.Walter, M. Döring, “Synthesis and Properties of Flame-retardant Epoxy Resins Based on DOPO and One of Its Analog DPPO”, J. Appl. Polym. Sci., 2007, 105, 685-696.

Author Response

References have been checked. Reference 23 has been corrected.

Reviewer 2 Report

Please refer Editors.

Author Response

Responses to comments have been provided in the initial response. No additional comments are available. The manuscript has been rechecked and seems suitable for publication (two reviewers have recommended publication).